

# Water deprivation-induced hypoxia and oxidative stress physiology responses in respiratory organs of the Indian stinging fish in near coastal zones

Samar Gourav Pati[1], Falguni Panda[1], Abhipsa Bal[1,2], Biswaranjan Paital[1] and Dipak Kumar Sahoo[3]

[1] Redox Regulation Laboratory, Department of Zoology, College of Basic Science and Humanities, Odisha University of Agriculture and Technology, Bhubaneswar, Odisha, India
[2] Department of Zoology, Regional Institute of Education, Bhubaneswar, Odisha, India
[3] Department of Veterinary Clinical Sciences, College of Veterinary Medicine, Iowa State University, Ames, IA, United States of America

Corresponding author
Biswaranjan Paital,
brpaital@ouat.ac.in,
biswaranjanpaital@gmail.com

## ABSTRACT

**Background**. Water deprivation-induced hypoxia stress (WDIHS) has been extensively investigated in numerous fish species due to their adaptation with accessory respiratory organs to respire air but this has not been studied in Indian stinging fish *Heteropneustes fossilis*. Data regarding WDIHS-induced metabolism in accessory respiratory organ (ARO) and gills and its relationship with oxidative stress (OS) in respiratory organs of air-breathing fish *H. fossilis*, are limited. So, this study aimed to investigate the effects of WDIHS (0, 3, 6, 12, and 18 h) on hydrogen peroxide ($H_2O_2$) as reactive oxygen species (ROS), OS, redox regulatory enzymes, and electron transport enzymes (ETC) in ARO and gills of *H. fossilis*.

**Methods**. Fish were exposed to air for different hours (up to 18 h) against an appropriate control, and ARO and gills were sampled. The levels of oxygen saturation in the body of the fish were assessed at various intervals during exposure to air. Protein carbonylation (PC) and thiobarbituric acid reactive substances (TBARS) were used as OS markers, $H_2O_2$ as ROS marker, and various enzymatic activities of superoxide dismutase (SOD), catalase (CAT), glutathione peroxidase (GPx), glutathione reductase (GR), along with the assessment of complex enzymes (I, II, III, and V) as well as the levels of ascorbic acid (AA) and the reduced glutathione (GSH) were quantified in both the tissues.

**Results**. Discriminant function analyses indicate a clear separation of the variables as a function of the studied parameters. The gills exhibited higher levels of GSH and $H_2O_2$ compared to ARO, while ARO showed elevated levels of PC, TBARS, AA, SOD, CAT, and GPx activities compared to the gills. The activities of GR and ETC enzymes exhibited similar levels in both the respiratory organs, namely the gills, and ARO. These organs experienced OS due to increased $H_2O_2$, TBARS, and PC levels, as observed during WDIHS. Under WDIHS conditions, the activity/level of CAT, GPx, GR, and GSH decreased in ARO, while SOD activity, along with GR, GSH, and AA levels decreased in gills. However, the activity/level of SOD and AA in ARO and CAT in gills was elevated under WDIHS. Complex II exhibited a positive correlation with WDIHS, while the other ETC enzymes (complex I, III, and V) activities had negative correlations with the WDIHS.

**Discussion**. The finding suggests that ARO is more susceptible to OS than gills under WDIHS. Despite both organs employ distinct redox regulatory systems to counteract this stress, their effectiveness is hampered by the inadequacy of small redox regulatory molecules and the compromised activity of the ETC, impeding their ability to effectively alleviate the stress induced by the water-deprivation condition.

## INTRODUCTION

The occurrence of oxygen ($O_2$) level fluctuation in aquatic environments is a common phenomenon (*Null, Mouzon & Elmore, 2017*). Under such circumstances, the limited or complete absence of $O_2$ supply gives rise to hypoxia or anoxia, respectively. Specific aquatic taxa, for example, fish, frequently encounter exceptionally low $O_2$ levels in the water as a consequence of seasonal habitat desiccation (*i.e.,* inhabiting mud flats during the summer) or diurnal water currents (*i.e.,* under tide and shoreline influx scenarios). As an illustration, catfish species like *Clarias* sp. and *Heteropneustes* sp. experience hypoxia during their annual inhabitation of mud flats in tropical countries. This phenomenon occurs due to the seasonal drying of ponds, ditches, rivers, and tributaries during summer. These environmental conditions can result in a nearly threefold elevation in oxidative stress (OS), notably measured through indicators such as the level of thiobarbituric acid reactive substances (TBARS) and protein carbonylation (PC) level in their organs (*Storey & Storey, 2005*; *Pelster et al., 2018*). In response to these challenging environmental circumstances, organisms have evolved with adaptations which include the ability of transitioning their respiration from water to air, utilizing accessory respiratory organ (ARO), skin and other mechanisms. These adaptive mechanisms serve to offset the extremely limited supply or absence of $O_2$ underwater by allowing these fish to respire $O_2$ from the air through their ARO, contrary to their usual reliance on gills for $O_2$ uptake when submerged (*Graham, 1983*; *da Cruz et al., 2013*).

The shift from an aquatic to an arid environment exacerbates the reduction in $O_2$ uptake through gills, while direct air-breathing for a prolonged period reduces mucin secretion in the ARO increasing the possibility of OS in fish (*Smatresk, 1986*; *Milsom, 2012*). The teleost catfish, *H. fossilis,* has evolved a specialized respiratory mechanism with the help of ARO, enabling it to endure water-deprived conditions for 60–70 h. Thus, this fish is preferably used as a model organism for conducting physiological studies related to stress in the context of hypoxia- induced by water deprivation conditions (*Olson et al., 1990*; *Panda et al., 2021a*; *Kumar, Gopesh & Sundaram, 2021*). Nevertheless, the physiological responses related to the OS in both the respiratory organs in the context of water deprivation-induced hypoxia stress (WDIHS) have not been investigated thus far (*Paital, 2013*; *Paital, 2014*; *Bal et al., 2022*). On the other hand, H. *fossilis* is being suggested as a potential model fish for coastal regions in several countries, including India, due to its notable capacity for

tolerance to high salinity levels (*Bal et al., 2021a*). In our previous work, we documented the ability of the fish to survive when directly exposed to water-deprived conditions for a few days. During this period, it was observed that the oxygen supply through the ARO in a water-deprived state might be insufficient to support the physiological process of the fish. This condition could potentially lead to hypoxia and consequently disrupt the equilibrium associated with OS in the respiratory organs (ARO and gills) of the fish (*Romero, Ansaldo & Lovrich, 2007*). This is attributed to the fact that OS and the regulation of redox processes are fundamentally reliant on oxygen-sensing bases for metabolic activities in animal cells. Moreover, their regulation is closely intertwined with the functioning of the electron transport chain (ETC) and oxidative phosphorylation (*Tisdale, 1967*).

Under non-stressful cellular conditions, mitochondria are the primary consumers of $O_2$, accounting for more than 90% of its utilization. Of this $O_2$ consumption, approximately 2–5% is responsible for generating the univalent reduced toxic superoxide anion radical. This radical subsequently gives rise to the highly toxic hydroxyl radical and hydrogen peroxide ($H_2O_2$) (*Abele & Puntarulo, 2004*; *Zorov, Juhaszova & Sollott, 2014*). To compensate for cellular energy homeostasis under a stressed state, mitochondria and their complex enzymes play an important role. As a result of this upregulation, the transport of electrons is accelerated leading to increased proton gradient. Hence, ATP is generated through oxidative phosphorylation (*Bertram et al., 2006*; *Bal et al., 2021a*). This phenomenon results in the leakage of electrons to reduce $O_2$ (a terminal electron acceptor) at complex I and III, giving rise to the above oxidants/radicals, collectively called reactive oxygen species (ROS) (*Liu, Fiskum & Schubert, 2002*; *Turrens, 2003*; *Zorov, Juhaszova & Sollott, 2014*). Therefore it is imperative to investigate the performance of ETC enzymes in the context of WDIHS for a comprehensive understanding of OS physiology (*Romero, Ansaldo & Lovrich, 2007*; *Murphy, 2008*). Consequently, this process triggers the involvement of the antioxidant system, including enzymes like superoxide dismutase (SOD, responsible for addressing superoxide radicals), catalase (CAT, responsible for managing $H_2O_2$) glutathione peroxidase (GPx, tasked with handling $H_2O_2$ and other organic hydroperoxides), glutathione reductase (GR, which recycles oxidized glutathione), ascorbic acid (AA, which acts non-specifically on ROS reduced glutathione (GSH, which acts non-specifically on ROS and recycles oxidized glutathione). These components work in concert to counterbalance the excessive production of ROS (*Limón-Pacheco & Gonsebatt, 2009*; *Idelchik et al., 2017*; *Panda et al., 2021b*; *Bal et al., 2021b*).

In specimens of *H. fossilis* subjected to water deprivation, a 23–25% increase in TBARS and $H_2O_2$ levels coupled with a significant reduction of 23% and 30% in the activity of antioxidant enzymes and ETC enzymes in its brain, respectively (*Paital, 2013*). Under analogous conditions, a downregulation of enzyme activity was observed in the fish after exposure to air, including a 55% decrease in SOD activity, a 228% increase in CAT activity, a 39% decrease in GPx activity and a 67% decrease in GR activity (*Paital, 2014*). In contrast to the brain, the liver exhibited an elevation in small antioxidant molecules and glutathione system by 15 to 20% under water deprivation for up to 6 h (*Bal et al., 2022*). The fish *H. fossilis* contains two branched gills on the lateral side of its head in the gill chamber (*Goel, 1978*; *Ratmuangkhwang, Musikasinthorn & Kumazawa, 2014*). Gills are lined with

epithelial cells in the form of branchial arches and gill filaments (*Mishra et al., 2011*; *Dey et al., 2015*). While submerged in water, this species depends on the lamella for respiration (*Bano & Hasan, 1990*) but, when exposed to air, they use two long tubular pneumatic sacs, located below the lateral line, for breathing. This organ extends from the branchial chamber to the tail region (*Goel, 1978*; *Ratmuangkhwang, Musikasinthorn & Kumazawa, 2014*). Interestingly, the OS status of organs directly involved in the breathing processes, such as gills and ARO, is yet to be explored under water-deprived conditions in this fish model (*Paital, 2013*; *Paital, 2014*).

During the conditions of water-deprived stress or low $O_2$ availability in water, the inflow of $O_2$ into the body and availability in cells decreases, which is usually associated with hypoxia in hypoxia (*Hermes-Lima et al., 2015*; *Liu et al., 2020*). The intricacies of ROS metabolism within aquatic organisms, including fish subjected to water deprivation, have yet to be comprehensively elucidated regarding their physiological processes (*Pelster et al., 2018*; *Giraud-Billoud et al., 2019*). Hence, we formulated a hypothesis suggesting that ARO and gills employ distinct strategies to maintain OS homeostasis when subjected to an air-breathing condition (Romeo et al., 2007; (*Paital, 2013*; *Paital, 2014*). To validate the hypothesis, we conducted a comprehensive examination of various parameters, including OS markers (TBARS, PC), ROS accumulation ($H_2O_2$), and activities of ETC (complex enzyme I, II, III, and V) as well as redox regulatory enzymes (such as SOD, CAT, GPx, and GR). We have also determined the levels of small antioxidant molecules (GSH and AA) in organs, gills and ARO of *H. fossilis* under water-deprivation stress (WDS). Furthermore, our study delved into the unexplored territory of the role played by complex II and other respiratory enzymes detected in gill and ARO of air-breathing fish *H. fossilis* under water deprivation stress. This research marks the first investigation of its kind. The analysis of physiological responses encompassing the aforementioned molecules in gills and ARO of the fish model *H. fossilis* exposed to aerial conditions-induced hypoxia offers valuable insights into our understanding of OS responses (*Zorov, Juhaszova & Sollott, 2014*).

## MATERIAL AND METHODS

All the analytical grade chemicals used in this experiment were purchased from Sigma Aldrich, USA, Merck, Germany, and Himedia and SD Fine Chemicals, Mumbai, India.

### Animal sampling, study design and organ collection

The rules and regulations of the Institutional Animal Ethical Committee of Odisha University of Agriculture and Technology were followed to handle the fish. Considering that the fish utilized in this experiment are not deemed vulnerable or at risk and are abundantly accessible, extensively consumed, and readily obtainable in the marketplace, there was no necessity to seek ethical approval for the execution of this study. However, the rules and regulations of the Institutional Animal Ethical Committee of Odisha University of Agriculture and Technology were strictly followed to minimize the stress on the fish and the number of animals required per experiment/group.

### Sampling of experimental fish

The fish *H. fossilis* ($n = 150$, $52 \pm 1.26$ g, $22.53 \pm 3.62$ cm) were collected using nets from a water body (pond) with limited anthropogenic activities (*Paital et al., 2018*) located at Anantapur, Machhagaon, Balikuda, Jagatsinghpur district, Odisha, India (20°03′37.0″N 86°20′24.0″E). The fish were captured with cast nets, immediately placed in an opaque plastic box supplied with ambient temperature and water, constantly aerated to minimize stress, and then transported to the laboratory under constant aeration. They were disinfected with 1–1,000 w/v potassium permanganate ($KMnO_4$) solution for 30 s, rinsed in ambient water, and then acclimatized to laboratory conditions (temperature $28 \pm 2$ °C, photoperiod 12 h: 12 h L: D) for 30 days in opaque plastic tanks of $1.5 \times 1 \times 0.5$ m$^3$ (*Paital & Chainy, 2013*; *Paital & Chainy, 2014*). The fish tanks (the fish number was limited to $n = 20$ to avoid crowding stress) with 100 L water were regularly aerated during the acclimatisation period to minimize stress and suffering to maintain the optimum dissolved oxygen (DO) level. They were fed with minced fresh chicken liver daily, and left-out feeds were cleaned after 1 h of feeding. The acclimatization of fish was routinely monitored, and feeding was given every morning, as previously described by *Bal et al. (2022)*. The mortality rate was recorded during the initial acclimatisation period, but no mortality rate was observed after a week.

### Experimental design

Based on previous experiments and the water deprivation withstand capacity of the fish *H. fossilis* up to 24 h, four sub-lethal experimental groups were set up against a control group, each having 10 ($n$) healthy and active fish. Therefore, we had ten biological replicates in each group. The actively swimming, feed-capturing fish were chosen for the study, while those with weak movement were not considered and released to their natural environment. For air exposure experiments, the fish were kept in opaque plastic trays under a water deprivation state for 3, 6, 12, and 18 h with appropriate control, *i.e.,* with 0 h exposure. So, the total number of fish used in this experiment was 50, *i.e.,* 10 in each group. Fish were sampled and assigned randomly into each group. The sample size was decided as per our previous experiment (*Bal et al., 2022*). To avoid over-drying the fish skin, water was sprayed at regular intervals of 12 h. Food was withdrawn 24 h before the set-up of the experiment. The experiments were performed with no sex differentiation.

### Measurement of O$_2$ saturation of fish under aerial exposure stress

A digital pulse oximeter (Model Black BPL Smart Oxy; Pulse Oximeter, Bengaluru, India) was used to measure the $O_2$ saturation level in the fish body at different intervals of water deprivation. The fish caudal part was inserted into the oximeter, and the fish was allowed to stay still. The readings were recorded in the fish at 0, 3, 6, 12, and 18 h of water deprivation. The readings were clear, non-fluctuating, and stable when measured in triplicates (Fig. S1).

### Collection of organ

After the water deprivation period, the fish ($n = 10$) were sacrificed following decapitation. Gills and ARO were excised, washed in saline solution (0.85% NaCl, w/v), and soaked using blotting paper, and flash-frozen in liquid nitrogen for further processing. Gills and

ARO 10% ($w/v$) organ homogenates were prepared in a homogenizing buffer (50 mM phosphate buffer pH 7.4 containing 330 mM sucrose and 1 mM phenylmethylsulfonyl fluoride as anti-protease). As described by *Bal et al. (2022)*, the crude homogenate was centrifuged at 4 °C to isolate the post-nuclear fraction (PNF), post-mitochondrial fraction (PMF), and mitochondrial fraction (MF). As described in previous works such as *Paital & Chainy (2012)*; *Paital & Chainy (2014)*, mitochondrial fractions were isolated (as seen by the Oxygraph study), and the PMF has no SDH activity, justifying that the followed procedure has good fractionation results for separating the PNF, PMF, and MF. The protein content in the above fractions was estimated using the Lowry method (*Lowry et al., 1951*).

### Determination of OS and ROS indices

The PNF was collected from organs and was used to determine the TBARS (nmol mg$^{-1}$ protein as per *Ohkawa, Ohishi & Yagi, 1979*) and PC (nmol g$^{-1}$ wet organ, as per (*Levine et al., 1994*), as markers of OS indices and H$_2$O$_2$ (ng g$^{-1}$ wet organ, (*Staniek & Nohl, 1999*), modified by (*Paital & Chainy, 2010*) as an index of ROS.

### Enzymatic antioxidants assays

The extracted PMF was used to measure the activities of redox regulatory enzymes at 25 °C, as described earlier (*Paital & Chainy, 2010*). The activity of SOD (EC1.15.1.1, unit mg$^{-1}$ protein, (*Das, Samanta & Chainy, 2000*), CAT (EC1.11.1.6, nano Kat mg$^{-1}$ protein, *Cohen, Kim & Ogwu, 1996*; *Aebi, 1974*), GPx (EC1.11.1.9, *Paglia & Valentine, 1967*) and GR (EC1.6.4.2, *Massey & Williams, 1965*) as unit of nmol of NADPH oxidized min$^{-1}$ mg$^{-1}$ protein was measured and calculated as mentioned in *Paital & Chainy (2010)*.

### Assay of non-enzymatic antioxidants

The PNF samples were precipitated in trichloroacetic acid (5% w/v) and centrifuged (10,000× g, 15 min) to obtain the clear supernatant. Non-protein sulfhydryl groups (*Sedlak & Lindsay, 1968*) and AA (*Mitsui & Ohta, 1961*) were measured in the obtained supernatant and were expressed as ng g$^{-1}$ of wet organ.

### Mitochondrial complex enzyme assay

After sonication of the mitochondrial ($n = 5$ in duplicates) fractions, the protein concentration was adjusted to ~1 mg of protein in 100 μL volume with appropriate dilution with the homogenizing buffer. In the sonicated samples, succinate dehydrogenase assay was conducted to check the purity of the mitochondrial fraction. The activities of complex enzymes such as I, II, III, and V were measured following the protocols given by *Gassner et al., (1997)*, *Lambowitz (1979)*, *Tisdale (1967)*, *Chen, Toribara & Warner (1956)* and *Cormier et al. (2001)*, respectively. The detailed protocols are referred to by *Paital & Chainy (2013)*.

### Statistical analyses

The data were examined for homogeneity of variance and normal distribution. The statistical analysis was conducted as per the data collected from the respective replicates ($n = 10$ for oxidative stress parameters: OSP, $n = 5$ for the ETC enzymes, and for O$_2$ saturation study in the fish body, where "n" denotes the number of animals in each set

of experimental groups). The same fish group was used to collect the data. The obtained values were represented as mean $\pm$ standard error of the mean. Correlation coefficient (r-value) values between the duration of water deprivation and OSPs were computed using Microsoft Excel version 10.0. Discriminant function analysis (DFA) was done as described earlier (*Paital, 2013*; *Paital, 2014*). The correlation coefficient observed from the DFA test was tabulated to know the contribution of variables to discriminate the group. Two-way ANOVA was performed to analyse the difference among the means values using SPSS statistics version 20. The evaluation of significance between two mean values was ascertained by Duncan's new multiple-range test. The differences between the computed data sets from each experimental group were considered statistically significant at $p < 0.05$.

## RESULTS

### Aerial exposure-induced hypoxia

Water deprivation induced hypoxia in the body of the fish, as evidenced by a decrease in $O_2$ saturation levels (Fig. 1). The decreasing trend between the duration of water deprivation and $O_2$ saturation has an "r" value of $-0.988$. This saturation level decreased significantly ($p < 0.003$) by 13.8%, 24.08%, 34.83%, and 51.98% at 3, 6, 12, and 18 h water deprivation states respectively when compared to the control group. Similarly, during the transitions from 0 to 3 h, 3 to 6 h, 6 to 12 h, and 12 to 18 h exposure to air, the $O_2$ saturation levels decreased significantly ($p < 0.01$) by 13.8%, 11.93%, 14.16%, and 26.31%, respectively, indicating a gradual decline in level of $O_2$ in the body of fish under air exposure state (Fig. 1).

### Variation in OS and ROS levels under aerial exposure stress

The assessed OS indices exhibited significant elevation in the studied organs following the water-deprivation state (Table 1). Notably, a time-dependent elevation in TBARS level was observed in ARO throughout aerial exposure. Specifically, the TBARS levels was found to be significantly higher ($p < 0.01$) by 8.6%, 23.7%, 45.2%, and 42.7% at 3, 6, 12, and 18 h of exposure, respectively, compared to the control set.

However, a distinct pattern was observed in the gills as compared to ARO. There was an increase in TBARS level in gills at all times of water-deprivation condition, with a particularly pronounced surge observed at 3 h. Specifically, TBARS levels were significantly higher ($p < 0.02$) 50.3%, 18.4%, 17.3%, and 18.4% higher at 3, 6, 12, and 18 h of exposure time compared to the control in gills.

Concurrently, the PC levels in gills displayed a similar trend, with a prominent spike at 3 h exposure time, recording a significant increase of 102.8% ($p < 0.001$) compared to the control. Subsequently, PC levels decreased ($p < 0.01$) by approximately 48.1%, 41.5%, and 39.6% at 6, 12, and 18 h, respectively, compared to the control set. It is noteworthy that the PC value in ARO exhibited a further increase during the water deprivation period of 18 h.

The concentration of $H_2O_2$ exhibited a gradual increase with each successive water-deprivation period in ARO. Specifically, the $H_2O_2$ concentration was recorded to be significantly higher ($p \leq 0.01$), 17.1%, 39.6%, 72%, and 109% at water deprivation durations of 3, 6, 12, and 18 h, respectively, in comparison to the control group in ARO.

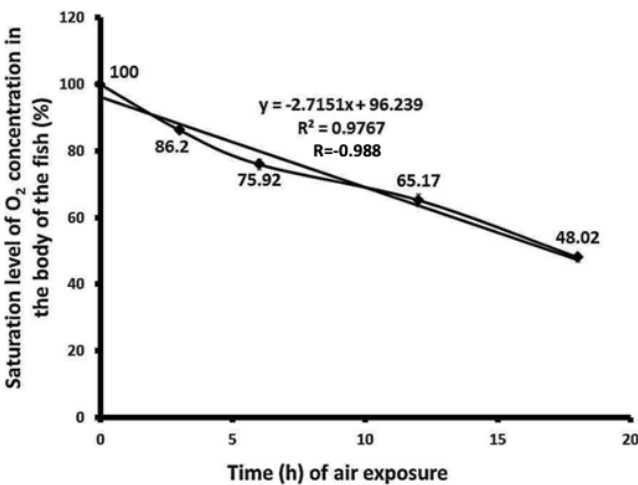

**Figure 1** **Water deprivation induced changes in O$_2$ saturation level in the catfish *Heteropneustes fossilis*.** The fish were exposed to aerial exposure from 0 to 18 h, and the O$_2$ saturation was measured at each interval. The data obtained at 0 h was considered an absolute scale to 100% saturation. Accordingly, the other values were converted to the subsequent O$_2$ saturation level compared to the control. The values (%) shown between each histogram indicate the percentage decrease in the immediate group from that of the previous group. Each histogram indicates the O$_2$ saturation point of the fish in the group. The error bars also indicate the variation Each original data set were compared by one-way ANOVA and data were presented as mean ($n = 3$) and standard deviation after one-way ANOVA analyses. The percentage values presented in between each histogram indicate the level of decrease of O$_2$ level from one ot another from lower to higher degree of air exposure time period. Any two sets of mean values were considered as significance at $p < 0.05$ level.

**Table 1** **Water deprivation induced hypoxia-mediated changes in oxidative stress and reactive oxygen species indices in the gill (G) and accessory respiratory organ (ARO) of the catfish *Heteropneustes fossilis*.** A study of up to 18 h water deprivation induced hypoxia-mediated stress was responsible for inducing ROS generation in the form of H$_2$O$_2$ in the fish, resulting in elevated lipid peroxidation in its gill and ARO. However, the ROS level was higher in gills than in ARO. LPx is presented as nmol of TBARS mg-1 protein, PC as nmol protein carbonyl g-1 wet organ and H$_2$O$_2$ as ng g-1 wet organ. Mean values (±SEM) with different superscripts are statistically different at $p < 0.05$ level, $n = 10$.

| OS parameters | Tissue | 0 h | 3 h | 6 h | 12 h | 18 h |
|---|---|---|---|---|---|---|
| TBARS | ARO | 15.60 ± 1.73[a,b] | 16.95 ± 2.64[a,b] | 19.31 ± 1.65[d] | 22.66 ± 1.34[e] | 22.27 ± 0.78[e] |
| | G | 14.71 ± 1.28[a] | 22.12 ± 0.86[e] | 17.42 ± 1.27[b,c] | 17.26 ± 1.46[b,c] | 17.41 ± 0.98[b,c] |
| PC | ARO | 3.69 ± 0.45[e] | 3.97 ± 0.3[f] | 3.60 ± 0.36[e] | 4.73 ± 0.36[g] | 4.88 ± 0.17[g] |
| | G | 1.06 ± 0.23[a] | 2.15 ± 0.19[d] | 1.57 ± 0.15[b,c] | 1.5 ± 0.11[b,c] | 1.48 ± 0.11[b,c] |
| H$_2$O$_2$ | ARO | 1.11 ± 0.23[a] | 1.3 ± 0.21[a,b] | 1.55 ± 0.11[b] | 1.91 ± 0.19[c] | 2.32 ± 0.15[d] |
| | G | 3.19 ± 0.33[e] | 5.04 ± 0.6[g] | 4.25 ± 0.24[f] | 4.54 ± 0.42[f] | 4.00 ± 0.35[f] |

Similarly, in gills, the quantity of H$_2$O$_2$ displayed a substantial increase ($p < 0.001$) at all assessed time points of water-deprivation conditions, reaching its peak with a 58% increase after 3 h of aerial exposure stress. Interestingly, following this initial rise, H$_2$O$_2$ levels in the gills declined and remained stable ($p < 0.78$) throughout the subsequent 18 h of exposure time (Table 1).

**Table 2** **Responses of redox regulatory and associated enzymes in the gill (G) and accessory respiratory organ (ARO) of the catfish *Heteropneustes fossilis* under the water deprivation induced hypoxia.** The ARO and gill tissues had different activities of the studied enzymes, such as superoxide dismutase (SOD as unit mg-1 protein, catalase (CAT as nano Kat mg-1 protein), glutathione peroxidase (GPx as nmol of NADPH oxidized min-1 mg-1 protein) and glutathione reductase (GR as nmol of NADPH oxidized min-1 mg-1 protein) under water deprivation for up to 18 h. Data represent the mean values (±SEM) of $n = 10$ samples, and the superscripts indicate statistical significance within the mean values at $P < 0.05$ level.

| Antioxidant enzyme | Tissue | 0 0 h | 0 3 h | 0 6 h | 0 12 h | 0 18 h |
|---|---|---|---|---|---|---|
| SOD | ARO | 0.66 ± 0.13[f] | 1.07 ± 0.2[g] | 1.68 ± 0.38[i,j] | 1.55 ± 0.31[h,i] | 1.33 ± 0.32[h] |
| | G | 0.33 ± 0.11[e] | 0.21 ± 0.02[d] | 0.12 ± 0.07[c] | 0.08 ± 0.03[b] | 0.03 ± 0.01[a] |
| CAT | ARO | 376.58 ± 15.9[f] | 314.36 ± 19.47[d,e] | 301.56 ± 45.9[d,e] | 306.55 ± 62.73[d,e] | 308.87 ± 44.04[d,e] |
| | G | 168.43 ± 14.34[b] | 135.89 ± 11.47[a] | 182.02 ± 19.9[b] | 219.14 ± 34.38[c] | 274.25 ± 65.96[d] |
| GPx | ARO | 95.86 ± 15.93[d] | 69.24 ± 7.72[c] | 65.19 ± 9.31[c] | 70.35 ± 16.18[c] | 70.07 ± 12.32[c] |
| | G | 27.54 ± 4.38[b] | 20.96 ± 3.56[a] | 18.38 ± 1.55[a] | 21.4 ± 4.01[a] | 28.85 ± 4.48[b] |
| GR | ARO | 2.33 ± 0.58[g] | 0.92 ± 0.2[d] | 0.89 ± 0.11[d] | 0.80 ± 0.12[b] | 0.84 ± 0.17[c] |
| | G | 2.56 ± 0.73[g] | 0.96 ± 0.14[e] | 0.96 ± 0.23[e] | 1.08 ± 0.14[f] | 0.33 ± 0.07[a] |

### Aerial exposure induced changes in activity of antioxidant enzymes

The alterations in redox regulatory enzymes induced by air exposure are detailed in Table 2. While the activity of GR exhibited similar levels in both organs, the activities of the other three redox regulatory enzymes, *i.e.,* SOD, CAT, and GPx, were remarkably higher in ARO as compared to the gills.

The activity of SOD demonstrated a consistent and significant increase ($p < 0.0001$) throughout all the durations of water-deprivation conditions in the ARO. The most substantial increase occurred at 6 h, reaching 154.5% above the control levels, and at the 12 h exposure point with a 134.8% increase (these two time points are statistically equivalent). Conversely, the SOD activity in the gills exhibited a consecutive reduction in response to the water-deprivation stress. Specifically, SOD activity in gills was notably lower ($p < 0.01$) by 36.4%, 63.7%, 75.8%, and 91% lower at water deprivation times of 3, 6, 12, and 18 h than the control set, respectively.

Water deprivation condition induced distinct alterations in the CAT activity in ARO and gills. CAT activity in the ARO gradually declined, reaching a 20% reduction compared to the control ($p < 0.05$) after 6 h of WDS. In contrast, the CAT activity in gills exhibited a progressive increase with the duration of water deprivation. Its peak activity, which was 14.6% higher than the control group ($p < 0.05$), was observed after 18 h of water deprivation, surpassing the other time points.

A gradual and significant decrease was noted in the GPx activity of ARO up to 6 h of WDS. Subsequently, its activity was slightly elevated at 12 and 18 h water-deprivation time points compared to the 6 h time point. However, there was a reduction in GPx activity between 3 and 12 h of water-deprivation in the gills, returning to the activity level of the control group after 18 h.

The enzymatic activity of GR exhibited noticeable variations in response to water-deprivation condition in both organs. GR activity declined significantly ($p < 0.01$) by 62% in the 6 h group in both organs compared to the control. In the ARO, the GR activity decreased up to 12 h of exposure to WDIHS and then slightly elevated at 18 h. However,

**Table 3  The role of the small antioxidant molecules, such as ascorbic acid (AA as ng g$^{-1}$ of wet organ) and reduced glutathione (GSH as ng g$^{-1}$ of wet organ), was evident in the studied tissues to protect them from oxidative stress under aerial exposure. Data are represented as mean values with a standard error of a mean of 10 fish.** Superscripts indicate statistical significance within the groups at $p < 0.05$ level.

| SAM | Tissue | 0 h | 3 h | 6 h | 12 h | 18 h |
|---|---|---|---|---|---|---|
| AA | ARO | 34.22 ± 7.86[c] | 38.92 ± 13.75[c] | 84.87 ± 17.96[d] | 89.84 ± 28.1[d] | 108.19 ± 20.98[e] |
| | G | 9.38 ± 2.42[b] | 10.90 ± 3.15[b] | 5.93 ± 3.05[a] | 8.14 ± 4.34[b] | 5.69 ± 2.88[a] |
| GSH | ARO | 78.84 ± 16.9[c,d] | 91.17 ± 5.98[d,e] | 69.41 ± 8.34[c] | 50.38 ± 5.44[b] | 29.07 ± 8.26[a] |
| | G | 178.39 ± 18.36[g] | 240.19 ± 19.02[h] | 84.63 ± 22.16[c,d] | 111.11 ± 12.47[e,f] | 118.27 ± 21.27[f] |

in gills, the GR activity steeply decreased up to 12 h, with a 57.8% lower activity ($p < 0.01$) at 12 h as compared to the control, but a sudden fall in the activity of GR was recorded at 18 h of WDIHS as compared to the other groups.

## Variation of small redox regulatory molecules

Water deprivation-induced changes in small redox regulatory molecules in ARO and gills are presented in Table 3. In ARO, the AA level steadily increased with respect to the time of water deprivation. In gills, the AA concentration remained stable at 3 and 12 h, and there was a reduction at 6 and 18 h in relation to the control. At 3 h of water deprivation, its level was 21.9% elevated ($p < 0.05$) compared to the control group, but at 18 h, it was 39.3% lower ($p < 0.01$) than the control.

The GSH-level in the ARO remained constant for up to 6 h and then declined for up to 18 h periods. In gills, the highest concentration of AA (52.6% higher, $p < 0.01$, than the control values) was recorded after 3 h of water deprivation, and then its level was reduced in the following period of water deprivation.

## Modulation of mitochondrial enzyme activity by water deprivation

All the mitochondrial complex enzymes exhibited reduced activities under the water deprivation state except complex II, which showed an increase in its activity with the proceeding time of water deprivation (Table 4). At 3, 6, 12, and 18 h of WDIHS, the activity of complex I enzyme was 24.8, 35.7, 53.4 and 50.4% lower ($p < 0.05$) in ARO and 15, 24.5, 41, and 50.8% lower ($p < 0.05$) in gills as compared to the control set, respectively.

However, water deprivation up to 3, 6, 12, and 18 h reduced ($p < 0.01$) the activity of complex III by 24.1, 32.1, 54.2 and 65.4% in ARO and 25, 40.6, 48.4 and 58.4% in gills as compared to the control group, respectively. Under 3, 6, 12, and 18 h of water deprivation, the activity of ATPase was reduced ($p < 0.01$) by 33.6, 76.1, 82.1, and 87.8% in ARO and 33.7, 76, 82, and 88% ($p < 0.01$) in gills, respectively, as compared to the control set. On the other hand, the activity of the complex II enzyme was stimulated ($p < 0.001$) by 44.4, 100, 111.1 and 133.3% in ARO and 28.6, 57.1, 100 and 128.6% in gills, as compared to the control group at 3, 6, 12, and 18 h of water deprivation periods respectively.

## Correlation and discriminant function analyses

A positive correlation was observed between all the studied OS and ROS indices (TBARS, PC, and H$_2$O$_2$) and the WDIHS in the ARO only (Table 5). Among the antioxidant enzymes, the activity of SOD was positively correlated with the water deprivation time,

**Table 4  Activity of respiratory chain enzymes in the gill (G) and accessory respiratory organ (ARO) of the catfish *Heteropneustes fossilis* under the water deprivation induced hypoxia.** The role of the mitochondrial complex enzymes was studied in G and ARO of the fish under aerial exposure stress. Data are represented as mean values with a standard error of a mean of 10 fish. The changes in the activities of each complex enzyme were evident from the study. Superscripts indicate statistical significance (two-way ANOVA) within the means values (±SEM) groups at $P < 0.05$ level.

| Complex enzyme | Tissue | 0 h | 3 h | 6 h | 12 h | 18 h |
|---|---|---|---|---|---|---|
| I | A | $2.38 \pm 0.25^c$ | $1.79 \pm 0.20^b$ | $1.53 \pm 0.37^b$ | $1.11 \pm 0.5^a$ | $1.18 \pm 0.97^a$ |
| | G | $3.27 \pm 0.25^e$ | $2.78 \pm 0.14^d$ | $2.47 \pm 0.40^c$ | $1.93 \pm 0.56^b$ | $1.61 \pm 0.12^b$ |
| II | A | $0.09 \pm 0.03^a$ | $0.13 \pm 0.02^b$ | $0.18 \pm 0.02^c$ | $0.19 \pm 0.03^c$ | $0.21 \pm 0.03^c$ |
| | G | $0.07 \pm 0.02^a$ | $0.09 \pm 0.02^a$ | $0.11 \pm 0.01^b$ | $0.14 \pm 0.02^b$ | $0.16 \pm 0.02^{b,c}$ |
| III | A | $16.74 \pm 2.88^e$ | $12.7 \pm 1.63^c$ | $11.37 \pm 3.09^c$ | $7.67 \pm 2.48^b$ | $5.80 \pm 0.76^a$ |
| | G | $18.74 \pm 2.88^e$ | $14.06 \pm 1.82^d$ | $11.13 \pm 1.97^c$ | $9.67 \pm 2.48^{b,c}$ | $7.80 \pm 0.76^a$ |
| V | A | $4.52 \pm 2.62^g$ | $3.00 \pm 1.12^e$ | $1.08 \pm 0.29^c$ | $0.81 \pm 0.30^{b,c}$ | $0.55 \pm 0.18^a$ |
| | G | $3.50 \pm 2.03^f$ | $2.32 \pm 0.86^d$ | $0.84 \pm 0.22^{b,c}$ | $0.63 \pm 0.23^b$ | $0.42 \pm 0.14^a$ |

**Table 5  Co-efficient (r) of correlation between oxidative stress physiology parameters along with the activity of respiratory chain enzymes and the time of water deprivation stress in *H. fossilis*.** The values of 'r' are given in the table with degrees of freedom as 49. Correlation coefficient (r) was taken as significant at a 5% level. 'ns' is assigned for non-significant correlation. A negative correlation was assigned with a negative sign with the r-value. Oxidative stress indices such as TBARS and protein carbonylation (PC), reactive oxygen species ($H_2O_2$), redox regulatory enzymes including superoxide dismutase (SOD, catalase (CAT), glutathione peroxidase (GPx) and glutathione reductase (GR), small antioxidant molecules namely ascorbic acid (AA) and the reduced glutathione (GSH) along with the complex enzymes (complex I, II, III, and IV) of respiratory chain were analyzed under the water deprivation conditions (0, 3, 6, 12 and 18 h) for the correlation study.

| Tissues | $H_2O_2$ | TBARS | PC | SOD | CAT | GPx | GR | AA | GSH | I | II | III | V |
|---|---|---|---|---|---|---|---|---|---|---|---|---|---|
| **ARO** | 0.97 | 0.89 | 0.83 | 0.62 | −0.57 | −0.53 | −0.57 | 0.92 | −0.95 | −0.93 | 0.91 | −0.91 | −0.88 |
| **Gill** | ns | ns | ns | −0.92 | 0.84 | ns | −0.72 | −0.43 | −0.57 | −0.94 | 0.90 | −0.86 | −0.87 |

while it was negatively correlated with the activity of CAT, GPx, and GR in the ARO. In this organ, the level of ascorbic acid had a positive correlation with the time of water deprivation, but the level of GSH had a negative correlation with the time of water deprivation (Figs. S2 and 3).

However, in gills, the activity of SOD and GR was negatively correlated, while CAT activity was positively correlated with water deprivation time. It was observed that both the AA and GSH level were negatively correlated in gills with respect to the time of water deprivation (Figs. S2 and 3).

The complex II activity had a positive correlation with the time of water deprivation, while the activity of the rest of the complex enzymes had a negative correlation with aerial exposure time in both organs (Figs. S2 and 3). Discriminant function analysis data with a high value for the canonical coefficients suggests a clear separation of variables in different water deprivation periods in the ARO (Fig. 2, Table 6). However, distinct separation among 0–3 h and the rest of the water deprivation groups was observed among all the exposure periods in gills (Fig. 2, Table 7).

## DISCUSSION

In this study, we embarked on a novel investigation, delving into the intricate realm of OS physiology within gills and ARO of the air-breathing fish model, *H. fossilis* under

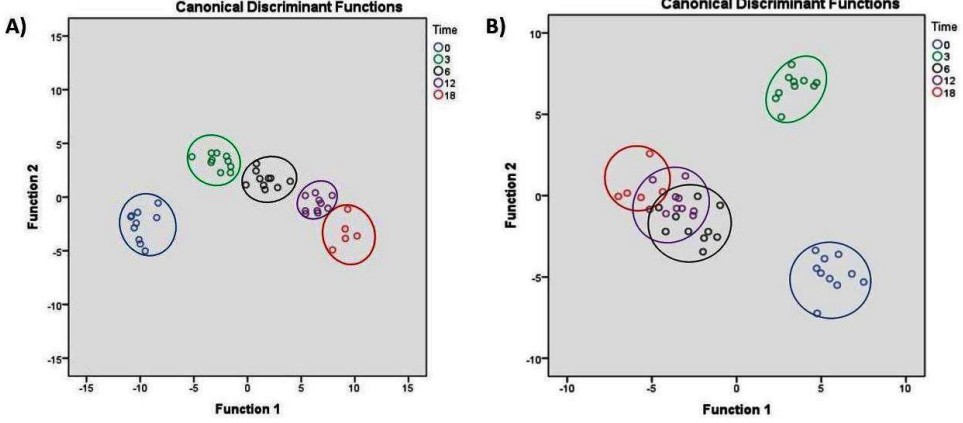

**Figure 2 Discriminant function analysis (DFA) on the studied oxidative stress physiology parameters as a function of water deprivation in *Heteropneustes fossilis*.** DFA is a statistical procedure used to determine the contribution of any particular variables in discriminating the groups. The pictorial view *via* different roots is considered to see visually whether the studied groups are separated into clear groups or not. The circular sets with smaller unicolor circles indicate each group as mentioned 0, 3, 6, 12, and 18 h aerial exposure groups. For example, the green circular set indicates in both the sub-s is 3 h aerial exposure group. The contribution of the studied variable can be known to form the canolical coefficient values as mentioned in Tables 6 and 7. Clear group discrimination was observed for all the parameters among 0, 3, 6, 12, and 18 h groups in ARO (A) than gills where overlapping among 0, 12, and 18 h groups was observed (B). All the studied parameters, such as oxidative stress indices, reactive oxygen species, antioxidant enzyme activities, redox regulatory molecules and complex enzyme activity levels in ARO (A) and gill (B) of *H. fossilis* as a function of water deprivation, were taken together for analysis.

the conditions of WDIHS. This exploration marked the first ever comparative analysis of how this particular fish species manages OS responses, with a particular focus on the ARO. Moreover, we ventured into uncharted territory by scrutinizing the role of complex II enzyme in the intricate web of OS physiology, a fact that has remained elusive in the understanding of animal physiology. Within this context, the generation of ROS through electron leakage at complex I and III enzymes has been well-established, leading to subsequent OS. However, the specific contributions of low oxygen content or hypoxia, have remained a mystery.

In the present study, we sought to illuminate this phenomenon by examining the responses in the catfish H. *fossilis* under the challenging conditions of WDIHS. Our research aims to shed light on the intricate mechanisms by which OS is managed in this unique aquatic species, particularly emphasizing the role of complex enzymes in ROS generation during periods of water deprivation conditions.

The catfish *H. fossilis* is proposed as a future model fish in many near coastal areas with lower salinity than seawater because this condition is prevalent in many countries, including India. After all, this fish has a salinity resistance capacity (*Bal et al., 2021a*). In air-breathing fishes such as *H. fossilis*, the presence of ARO plays an important role in extracting $O_2$ from the ambient air (*Pelster et al., 2018*). The presence of ARO is considered an evolutionary adaptation at the anatomical level in some aquatic animals, including

**Table 6 Standardized classification function coefficient for the canonical variables observed for the oxidative stress physiology parameters and the respiratory complex enzyme activities in the accessory respiratory organ of the teleost fish *Heteropneustes fossilis* as a function of aerial exposure stress.** All the studied parameters, such as TBARS, protein carbonylation (PC), reactive oxygen species ($H_2O_2$) redox regulatory enzymes, including superoxide dismutase (SOD, catalase (CAT), glutathione peroxidase (GPx) and glutathione reductase (GR), small antioxidant molecules namely ascorbic acid (AA) and the reduced glutathione (GSH) along with the complex enzymes (complex I, II, III, and IV) of respiratory chain were analyzed together for discriminant function analyses.

| Parameters | Time of water deprivation | | | | |
|---|---|---|---|---|---|
| | 0 hour | 3 hours | 6 hours | 12 hours | 18 hours |
| SOD | 26.268 | 43.048 | 61.085 | 62.943 | 62.281 |
| CAT | 0.181 | 0.140 | 0.158 | 0.161 | 0.159 |
| GPx | 0.617 | 0.125 | −.086 | −.246 | −.374 |
| GR | 9.986 | −25.389 | −24.002 | −32.021 | −32.238 |
| AA | 0.265 | 0.265 | 0.482 | 0.443 | 0.478 |
| GSH | 0.757 | 1.015 | .632 | 0.277 | -.087 |
| $H_2O_2$ | −7.159 | 20.527 | 35.247 | 57.554 | 77.414 |
| PC | 16.697 | 31.427 | 29.617 | 42.874 | 45.591 |
| TBARS | 4.310 | 4.953 | 6.017 | 7.258 | 7.208 |
| I | 17.883 | 16.827 | 18.191 | 15.345 | 16.036 |
| II | 594.099 | 632.788 | 731.940 | 659.968 | 608.862 |
| III | 6.582 | 5.658 | 6.115 | 4.754 | 3.848 |
| V | 1.539 | 2.308 | 1.171 | .810 | .372 |
| (Constant) | −288.210 | −304.145 | −358.228 | −409.350 | −426.544 |

fishes, to cope with periods of arid conditions, such as the dry seasons (*Damsgaard et al., 2020*). Many fish species have evolved ARO to compensate for limited $O_2$ supply *via* gills under water-deprivation conditions. Such frequent events are observed in freshwater fishes during dry seasons and marine fishes during tide and influx (*Storey & Storey, 2005*).

Despite the wealth of literature explaining hypoxia conditions in the body of fishes and the physiological responses of aquatic animals with respect to OS, reports are scanty in the context of gills and ARO in particular (*Welker et al., 2013*; *Hermes-Lima et al., 2015*), Table 8). It can be hypothesized that utilization of ARO for $O_2$ uptake may potentially exert a range of stressors in this organ which includes ROS, OS, haematological disturbances, high neural transmission, immune responses, and other stress-related factors like ammonia load (*Paital, 2013*; *Paital, 2014*; *Mishra & Mishra, 2017*; *Chew et al., 2020*; *Lauriano et al., 2021*). In light of these considerations, our study contributes valuable insights into the complex interplay between OS responses and the unique respiratory adapting mechanisms of *H. fossilis*.

The physiological disruptions induced by stress in fish species like *H. fossilis* are well documented (*Samim & Vaseem, 2021*). These disruptions can be attributed to the fact that certain organs, like the ARO, remain inactive for $O_2$ respiration when the fish thrives in water. Consequently, the patterns of redox regulatory responses in this organ may diverge from those observed in gills that are inactivated during the WDIHS period (Table 8). Thus, gaining insights into the OS physiology in both gills and ARO is significant for

Table 7 **Standardized classification function coefficient for the canonical variables observed for the level of oxidative stress parameters, reactive oxygen species, level of small antioxidant molecules, and for the activities of antioxidant and complex enzymes in the gill of the teleost fish *Heteropneustes fossilis* as a function of aerial exposure stress.** All the studied parameters, such as TBARS, protein carbonylation (PC), reactive oxygen species ($H_2O_2$) redox regulatory enzymes, including superoxide dismutase (SOD, catalase (CAT), glutathione peroxidase (GPx) and glutathione reductase (GR), small antioxidant molecules namely ascorbic acid (AA) and the reduced glutathione (GSH) along with the complex enzymes (complex I, II, III, and IV) of respiratory chain were analyzed together for discriminant function analyses.

| Parameters | Time | | | | |
|---|---|---|---|---|---|
| | 0 hour | 3 hours | 6 hours | 12 hours | 18 hours |
| SOD | 57.374 | −1.439 | 17.655 | −2.302 | −34.270 |
| CAT | 0.336 | 0.313 | 0.365 | 0.393 | 0.462 |
| GPx | 1.824 | 2.107 | 1.247 | 1.697 | 2.562 |
| GR | 34.183 | 18.858 | 23.393 | 23.636 | 10.934 |
| AA | 1.064 | 1.907 | 1.283 | 2.252 | 2.101 |
| GSH | 0.457 | 0.754 | 0.184 | 0.311 | 0.347 |
| $H_2O_2$ | 22.873 | 41.563 | 32.798 | 22.134 | 18.384 |
| PC | 22.155 | 34.304 | 30.649 | 34.773 | 31.460 |
| TBARS | 9.377 | 13.863 | 11.099 | 11.434 | 11.996 |
| I | 36.790 | 34.895 | 31.919 | 27.259 | 27.739 |
| II | 396.202 | 348.002 | 430.237 | 554.620 | 556.643 |
| III | 3.389 | 0.461 | 1.860 | 1.373 | 0.630 |
| V | −12.254 | −10.998 | −11.027 | −12.464 | −9.818 |
| (Constant) | −354.768 | −494.595 | −325.704 | −364.256 | −381.909 |

comprehending the role of hypoxia-induced due to air exposure and mechanisms associated with OS. Regrettably, comparative studies investigating the impact of water deprivation on OS in ARO and gills of fish, including *H. fossilis* are relatively scarce (*Graham, 1983*; *Smatresk, 1986*; *Olson et al., 1990*; *Storey & Storey, 2005*; *Milsom, 2012*; *da Cruz et al., 2013*; *Paital, 2013*; *Paital, 2014*; *Pelster et al., 2018*; *Kumar, Gopesh & Sundaram, 2021*); *Panda et al., 2021a*; *Bal et al., 2022*, Table 8). Therefore, the primary objective of this work was to study water-deprivation conditions induced hypoxia pattern, ROS generation, OS pattern, the responses of redox regulatory enzymes and non-enzymatic molecules, and the activities of ETC enzymes in gills and ARO of the catfish *H. fossilis*.

Under WDIHS, although ARO acts to compensate for the limited or no $O_2$ supply to the fish body through the gills, it was found insufficient in *H. fossilis*. It can be explained that water deprivation induced hypoxia in this fish, and the fish had about 94–96% $O_2$ saturation in their bodies at a 0 h air WDIHS. This result is in agreement with a previous study (*Bal et al., 2022*), which indicates that a severe interruption in $O_2$ supply to the body of the fish arises under WDIHS (*Paital, 2013*; *Paital, 2014*). It is pertinent that adaptation to air breathing nature of *H. fossilis* by ARO under aerial exposure state is a temporary contextual mechanism, and it is employed only to enable this fish to survive for a few hours to a maximum of three days under water deprivation state (*Saha & Ratha, 1989*; *Saha et al., 2001*).

**Table 8 Responses of redox regulatory enzymes in different fishes under hypoxia, hyperoxia and anoxia -induced stress.** The percentage of change was calculated with respect to their highest or lowest activity observed in any of the experimental group as compared to the control group in the respective fishes.

| Fish | Condition | Tissue | Variation of antioxidant system |
|---|---|---|---|
| *Perccottus glenii* (**The Chinese sleeper**) | Hypoxia ($0.4$ mg $O_2$ $L^{-1}$) (2, 6 and 10 h0 | Brain Liver Muscle | SOD (41.1%) ↓, GR (63.6%) ↓, SOD (200%) ↑, CAT (64.5%) ↓, GR (45%) ↓, SOD (138%) ↑ |
| *Cyprinus carpio* (**European carp**) | Hypoxia ($0.9$ mg $O_2$ $L^{-1}$) (5.5 h) | Brain Liver Muscle | CAT (57.1%) ↑ GPx (29%) ↑ SOD (50%) ↓ GPx (33.7%) ↓ GPx (30%) ↓ |
| *Piaractus mesopotamicus* (**Pacu**) | Hypoxia (50 mm Hg) (48 h) | Liver Muscle | CAT (29.7%) ↓ GPx (17.6%) ↓ CAT (54.1%) ↓ GPx (41.1%) ↓ SOD (60.1%) ↓ |
| *Cyprinus carpio* (**The Eurasian or European carp**) | Hypoxia (8 h) | Brain Gill Liver | SOD (78.6%) ↑ SOD (150%) ↑ SOD (500%) ↑ |
| *Gadus morhua* (**The Atlantic cod**) | Hypoxia (46% $O_2$ saturation) (6 weeks) | Liver | GPx (150%) ↑, SOD mRNA (50%) ↑ |
| *Sparus aurata* (**Orata in antiquity**) | Hypoxia ($2.8$ mg $O_2$ $L^{-1}$) (3 and 6 h) | Liver | GPx (16.6%) ↑, GSH (20%) ↓ |
| *Leporinus elongatus* (**Piapara**) | Hypoxia ($1.92$ mg $O_2$ $L^{-1}$) (14 days) | Blood Liver | GSH (95.6%) ↑, SOD (215.4%) ↑ SOD (44.6%) ↑, GPx (18%) ↑, GST (428.4%) ↑, GSH (37.5%) ↑ |
| *Leporinus macrocephalus* (**Piauçu**) | Hypoxia ($0.71$ mg $O_2$ $L^{-1}$) (4 days) | Liver | GST (43%) ↓, CAT (41.7%) ↓ |
| *Hyphessobrycon Callistus* (**The Red Minor tetra**) | Hypoxia ($1.0$ mg $O_2$ $L^{-1}$) (2.5 h) | Blood serum | SOD (27.6%) ↑, GPx (244%) ↑ |
| *Carassius auratus* (**The goldfish**) | Anoxia (8 h) | Brain Kidney Liver | GPx (86.6%) ↑, G6PDH (25.7%) ↑, CAT (10.8%) ↓, GPx (17.3% ↓ CAT (61.1%) ↑ |
| *Gadus morhua* (**The Atlantic cod**) | Hyperoxia (145% $O_2$ saturation) (6 weeks) | Liver | GPx mRNA (150%) ↑ |
| *Carassius auratus* (**The goldfish**) | Hyperoxia (18–20 mg $O_2$ $L^{-1}$) (3, 6 and 12 h) | Liver Kidney | GST (57.1%) ↓ G6PDH (60.1%) ↑ |
| *Salmo salar* (**The Atlantic salmon**) | Hyperoxia (140% $O_2$ saturation) (6 weeks) | Liver | SOD (8%) ↓ GPx (22.2%) ↓ |
| *Heteropneustes fossilis* (**Stinging catfish**) | Hypoxia (3, 6, 12 and 18 h) | Brain | SOD (76%) ↓, CAT (33.3%) ↑, GPx (57%) ↓, GR (67%) ↓ |
| *Heteropneustes fossilis* (**Stinging catfish**) | Hypoxia (3, 6, 12 and 18 h) | Liver | SOD (69.2%) ↓, CAT (39.1%) ↓, GPx (63%) ↑, GR(67%) ↓ |
| *Heteropneustes fossilis* (**Stinging catfish**) | Hypoxia (3, 6, 12 and 18 h) | Muscle | SOD (55%) ↓, CAT (228%) ↓, GPx (39%) ↓, GR (67%) ↓ |

The level of $O_2$ is also important to produce ROS and OS in any organ or tissue. Under WDIHS, the gills remain inactive as the water supply is restricted, and ARO has more access to $O_2$ as it respires air under WDIHS. Therefore, this could be a reason why ARO had a higher OS level than gills under WDIHS. At the same time, ARO maintains a high level of redox regulatory system, so the $H_2O_2$ level was less in this organ than gills. Even if blood was not fully saturated with oxygen, as hypoxia was observed in the fish during

water deprivation, air-exposed epithelial cells would face high $PO_2$ values. But it remained unanswered what was the $O_2$ diffusion rate to ARO epithelial cells and how much $O_2$ was available in this organ during water deprivation. Overall, since hypoxia was induced in the fish under WDIHS, it could be presumed that the generation of ROS and OS was due to the maximum reduction of mitochondria in both gills, and ARO could be low compared to when the fish normally live in water.

In contrast to the expectation, the OSPs and ROS indices (TBARS, PC, and $H_2O_2$) were positively correlated with the water deprivation time. It confirms that the elevation of ROS and oxidation of lipids and proteins in gills and ARO of *H. fossilis* is evident under WDIHS-induced hypoxia (*Paital & Chainy, 2010*). Earlier, organ-specific variation of TBARS in different animals, especially in fishes, was recorded under different hypoxia and reoxygenation states (*Lushchak et al., 2005*). Although the studied ROS level was lower in ARO than in gills, higher TBARS, and PC levels were observed in the former organ, which could be due to the differential engagements of both organs by the fish under a water deprivation state. It could be due to varied metabolic activities in gills and ARO as the former remains dormant and the latter remains most active to supply $O_2$ under a water deprivation state.

The observed low ROS levels in ARO than gills could be an organ-specific adaptation by the fish to withstand any possible higher OS generation under an aerial exposure-induced hypoxia state. It indicates that the ARO might follow the proposed theory of "preparation of oxidative stress" (*i.e.,* the organ accumulates higher ROS and the antioxidant defense as well under any preliminary state of the stress) underwater deprivation-induced hypoxia condition (*Giraud-Billoud et al., 2019*). The lowered PC level with increased TBARS value was previously observed in the liver of common carp, *Cyprinus carpio,* under five hours of hypoxic conditions (*Lushchak et al., 2005*). Some other studies related to animals that undergo aestivation or are exposed to $O_2$-compromised conditions have also shown less ROS production, as a protective mechanism against hypoxia. Such adaptation of low ROS production hypoxia or aestivation is observed in the African lungfish, *Protopterus annectensin* (*Loong et al., 2012*; *Ong et al., 2015*; *Ong et al., 2017*; *Hiong et al., 2015*), the alleviating *P. annectensin* (*Chng et al., 2014*), respectively.

Similarly, elevated TBARS and $H_2O_2$ levels with an unchanged PC value are documented in another catfish, *Clarias batrachus,* under lead exposure for up to 20 days (*Maiti, Saha & Paul, 2010*). In the current study, the critical period for the stress condition in *H. fossilis* was 3 to 6 h in the gills, whereas it was 3 to 12 h (3 h for PC and 12 h for TBARS) in ARO. This could be an indication of the evolutionary adaptation for the protection of ARO rather than gills from OS under WDIHS because ARO respires air and is recruited by the fish for survival under aerial exposure rather than gills. However, hypoxia-induced higher ROS generation may indicate the employment of the alternate oxidase system or back flow of electrons *via* ETC in the fish, and it needs further investigation for confirmation (*Turrens, 2003*).

The biochemical responses of redox regulatory enzymes and supporting enzymes such as GR under WDIHS shall provide a further understanding of the OS physiology in gills and ARO of *H. fossilis*. Elevation in the activity of GPx and CAT was observed in *C. carpio* under

air exposure for up to 5 h (*Lushchak et al., 2005*). In the gills of *Chasmagnathus granulata*, an alleviation of SOD activity and elevation of other antioxidant enzyme activities, such as CAT and GST, was documented (*de Oliveira et al., 2005*). On the other hand, *De Almeida & Bainy (2006)* observed that other than the activity of SOD, no other redox regulatory enzyme activity was elevated at the biochemical level in brown mussel, *Perna perna,* under air exposure state. The downregulation of activities of most of the antioxidant enzymes under WDIHS-induced hypoxia in the present study indicates the impairment of the antioxidant system in both the organs of *H. fossilis.* However, the activity of SOD in ARO and CAT in gills was up-regulated gradually with respect to the WDIHS period. In gills, the activity of SOD and GR was observed to be the lowest at 18 h of water deprivation. However, the activity of GPx was recovered at 18 h, and it was probably to reduce the overproduced ROS. No recovery in the activity of redox regulatory enzymes was observed in the ARO, for which the OS levels were high in this organ at 18 h WDIHS.

The SOD activity was found to be elevated, but on the contrary, the activity of both CAT and GPx enzymes was lowered in the ARO under WDIHS. Therefore, the elevation of the dismutation of superoxide anions to $H_2O_2$ and the reduction in the conversion rate of $H_2O_2$ to water molecules might be the major reasons for the observed higher levels of $H_2O_2$ WDIHS in the organ. The reason for a "good" protection against superoxide radicals, but a low defense against $H_2O_2$ needs further study. Similarly, the lower activity of SOD and higher activity of CAT were responsible for reducing $H_2O_2$ concentration in gills under WDIHS. The sudden surge in $H_2O_2$ level at 3 h WDIHS suggests the insufficient activity of CAT to diminish OS. After 6 h, the increase in the activity of CAT was observed to manage the $H_2O_2$ level in the gills. The activity of GPx enzyme was found to be decreased in both organs with respect to WDIHS. It might be due to less titer of the free GSH group used by this enzyme to discharge its activity (*Paital, 2013*). The lower activity of GR enzyme supports this fact as it regenerates GSH from its oxidised form. The activity of SOD and level of ROS were also reduced in the African lungfish, *P. annectens,* during aestivation, where $O_2$ uptake is expected to be low (*Hiong et al., 2015*). In contrast, in the same lungfish, the activity of mitochondrial and cytosolic SOD, CAT, GPx, and GR was upregulated in brain tissue after 60 days of aestivation (*Page et al., 2010*), indicating separate redox strategies maintenance in different fish under hypoxia state.

For the synthesis of AA, an enzyme called L-gulonolactone oxidase is necessary, which is absent in higher animals but is readily functional in most invertebrates and certain higher vertebrates (*Halliwell & Gutteridge, 2001*). This enzyme is present, and some evidence shows that this enzyme can be synthesized in some fishes during a time of need (*Moreau & Dabrowski, 2000*). Ascorbic acid in the ARO was observed to exhibit a progressive increase under the WDIHS. However, in gills, the elevation of ascorbic acid was comparatively lower (10.6%) after 3 h of exposure, as compared to the control group. This result explains the probable chance of internal synthesis of AA in *H. fossilis* as the fish were deprived of food supply prior to experiments (*Moreau & Dabrowski, 2000*). The higher level of AA served to reduce ROS levels in ARO to protect the organ from hypoxia. The qualitative determination of the presence of L-gulonolactone oxidase might explain the above fact. The fact of upregulation of L-gulono-7-lactone oxidase expression and synthesis of vitamin

C in the brain and kidney of the African lungfish, *P. annectens* under aestivation (that leads to hypoxia) has been noted (*Ching et al., 2014*). The GSH group level was found to be increased at 3 h of water deprivation time in both organs, and it was to counter the OS. However, after 3 h, its level declined further. The decrease in GR activity in both organs might be one of the causes of the lowered thiol group level at 3 h WDIHS because the enzyme GR regenerates GSH from its oxidised form GSSG (*Carlberg & Mannervik, 1985*).

The increase in the activity of GPx at 12 and 18 h compared to the previous time of exposure points was the reason for the observed lower level of GSH in these groups. Such data are supported by the findings of *Mansfield, Simon & Keith (2004)*, who recorded the gradual reduction of GSH group level in *different cell lines* under hypoxic conditions. In the case of small redox regulatory molecules, the critical time of hypoxia was observed to be 3 h of WDIHS.

The combined action of the ETC enzymes present in the inner membrane of mitochondria provides information about oxidative phosphorylation as well as the rate of synthesis of ATP (*Liu, Fiskum & Schubert, 2002*). All the complex enzymes, except the complex II enzyme, were found to be gradually downregulated under the WDIHS in *H. fossilis*. Out of all the respiratory enzymes, complex I and III are considered to make the highest contributions to ROS generation in the mitochondrial matrix as well as the inner mitochondrial space (*Halliwell & Gutteridge, 2001*) because the leakage of electrons to reduce $O_2$ takes place at the above complex enzyme. Activities of complex I and III were downregulated in the present experiment with respect to WDIHS, and it can be explained that the reduction or dysfunction of the activity of these complex enzymes could lead to the elevation of ROS generation in mitochondria due to electron leakage at these two enzymatic sites as compared to complex II (*Liu, Fiskum & Schubert, 2002*; *Paital, 2014*). Therefore, the lower activity of complexes I and III would lead to producing more ROS under WDIHS in both the organs of the fish. At the same time, the higher activity of complex II was probably to supply the required electrons to ETC. *Zaccone et al. (1985)* also found a similar reduction of respiratory enzymes in *H. fossilis* after exposure to the toxicant sodium alkylbenzene sulfonate for up to 24 h. After 20 days of lead exposure, $Na^+$-$K^+$ ATPase and complex V dysfunction were observed in the catfish *C. batrachus* (*Maiti, Saha & Paul, 2010*).

The activity of complex V is usually measured *in vitro* by ATPase activity rather than estimating ATP formation (*Paital, 2013*). The negative correlation of complex V with the water deprivation period suggests diminished ATP formation, which could lead to an imbalance between energy expenditure and demand or supply. However, the activity of complex II was increased under WDIHS, and it was probably to act as an adaptation to maintain electron transfer activity *via* ETC. It is because the electron entry into ETC occurs at two complex enzymes, *i.e.,* complex I and complex II. Considering the observed decrease in the activity of complex I during whole-cell respiration, it is plausible to posit that the concomitant increase in the activity of complex II serves as a compensatory mechanism to facilitate the entry of electrons into ETC. The critical time of complex enzymes to the hypoxia condition of complex V was recorded at 3 h of water deprivation, while this value for other complex enzymes was at 6 h of the water deprivation period. A similar reduction

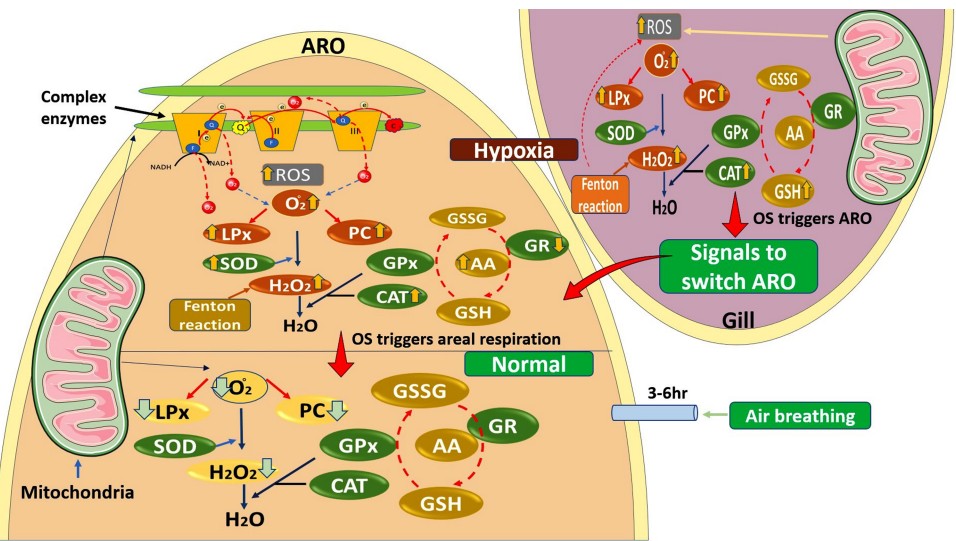

**Figure 3** **The possible mechanism of water deprivation -induced hypoxia-mediated modulation of electron transport chain enzymes, reactive oxygen species production, and other oxidative stress physiology parameters in gill and ARO organs of *H. fossilis*.** It was observed that both organs adapt different mechanisms for modulation of oxidative stress responses under aerial exposure-induced hypoxia stress in the fish. Hypoxia-induced oxidative stress leads to the formation of ROS and oxidation of protein, nucleic acids, and lipids forming protein carbonyls, nucleic acid adducts, and lipid peroxide. In mitochondria, complex enzymes play an important role in the formation of ROS due to the leakage of electrons at complex enzyme I and III. Small antioxidants like ascorbic acid and GSH lessen OS in both gill and ARO under low oxygen levels. $O_2^-$-superoxide radical, AA–ascorbic acid, ROS–reactive oxygen species, CAT–catalase, GPx–glutathione peroxidase, GSH- reduced glutathione, GSSG–oxidized glutathione.

(67%) in cytochrome C oxidase activity has been noticed in the estivating African lungfish *Protopterus dolloi* after six days of water deprivation (*Frick et al., 2010*).

Due to the inability to function as a major oxygen-respiring organ, ARO was unable to supply enough oxygen to the body of the fish *H. fossilis* under water deprivation, leading to hypoxia. The observed occurrence of OS in the gills and ARO of the fish can be attributed to the inadequate functioning of various redox regulatory enzymes and the ETC enzymes. The potential consequences of this phenomenon may manifest as harmful effects on the investigated organ, characterized by the accrual of TBARS and PC contents (Fig. 3). It was associated with the downregulation of the activity of complex V enzyme, which in turn could lead to less ATP molecule production in the organs. The diminished activity of ETC enzymes, especially complex I, III, and V, could be the cause for generating more ROS under WDIHS in the studied organs of the fish. The fish adapts different redox regulatory strategies to protect ARO than gills under WDIHS-induced hypoxia, ROS generation, and OS.

Results of the present study suggest the failure of the antioxidant system to counter the ROS and its effect after 3 to 6 h of water deprivation in ARO and gills of *H. fossilis*. The potential etiology underlying the heightened levels of ROS in the body of fish could be attributed to a diminished functionality of mitochondrial respiratory enzyme complexes and redox regulatory enzymes. The increase in the activity of complex II, in particular,

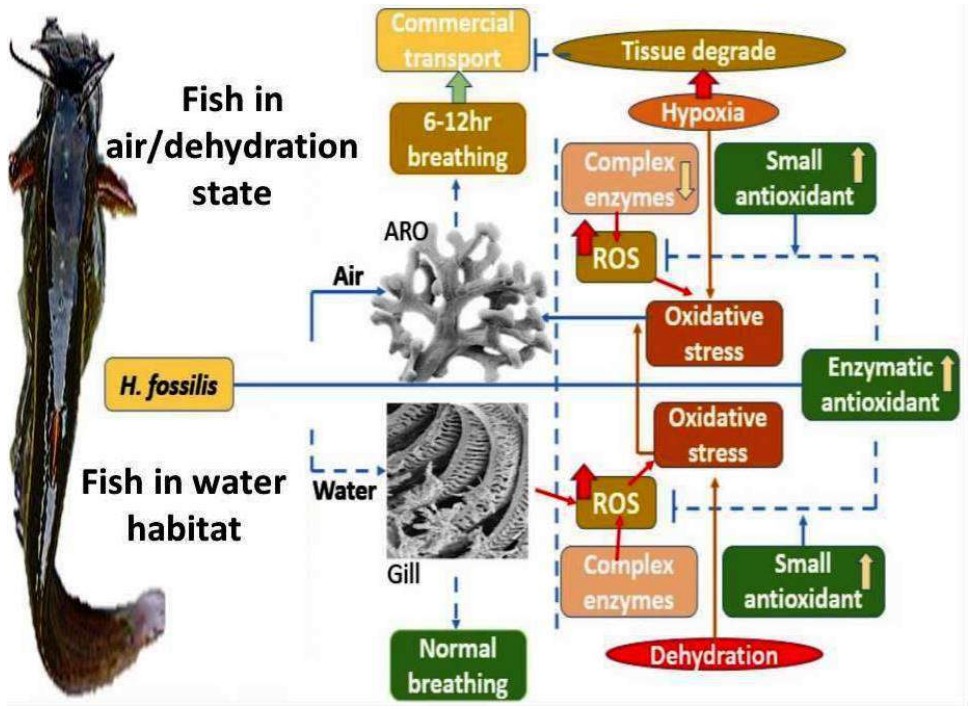

**Figure 4** **Differential variation of redox state enzyme and non-enzymes to generate oxidative stress in *H. fossilis* under water-deprived stress.** The "↓" symbol is used for the decrease in level, and the "↑" symbol is for an increase in the level of the respective parameters in accessory respiratory organs (ARO) and gills of *H. fossilis*. The variations of the studied parameters were differentially regulated, as shown in the figure. The protective action of the small redox state molecules, especially that of ascorbic acid, may indicate an evolutionary adaptation in the fish that needs further study. The photos of fish are taken with a mobile camera by the 3rd author.

provides an opportunity to investigate its role in increasing ROS generation in fish. Levels of TBARS and PC indicate decreased cellular activity, particularly the failure of the redox system in fish subjected to water deprivation-induced hypoxia. However, future studies on the recovery phase of the antioxidant system in *H. fossilis* will provide a better understanding of OS physiology in such fishes. The present study elucidates the mechanism of OS physiology under water deprivation-induced hypoxia in the respiratory organs of the fish. The role of AA in the ARO under hypoxia is an evolutionary adaptation by the fish to protect this organ from OS (Fig. 4). Such hypoxia-induced OS mechanisms have already been established in Characid fish *Cyphocharax abramoides* (*Aon, Cortassa & O'Rourke, 2010*; *Johannsson et al., 2018*).

From the correlation analyses between the patterns of the studied parameters (ROS and OS level, redox regulatory molecules, and ETC enzymes) and the water deprivation period, it is concluded that WDIHS-induced hypoxia modulates the above parameters in an organ-specific way. Mostly WDIHS-induced hypoxia regulates OS physiology more vividly in ARO than in gills as this organ is insufficient to compensate for limited or no O$_2$ supply. Except for the succinate dehydrogenase activity, the rest of the studied complex enzymes

in mitochondria in both organs were negatively correlated with WDIHS, indicating the reduced activity of ETC in mitochondria. Succinate dehydrogenase, an enzyme of utmost significance, fulfills its pivotal function in both the Krebs cycle and the ETC. Hence, the activity of this enzyme observed in the present study does not provide sufficient evidence to definitively determine whether it originated from the Krebs cycle or the ETC. However, results from DFA indicate a clear contribution of the studied variables for separating groups into five. Specifically, WDIHS had significant effects on ARO, as a clear separation of groups with high values of the canonical coefficient was observed when data for this organ was considered together for analysis. A distinct overlapping among 6, 12 and 18 h groups in gills indicates that the fish does not employ this organ for any metabolic activity under WDIHS as these parameters were clumped together. However, both correlation and DFA analyses indicate that WDIHS has a definite role in the studied parameters in gills and ARO of *H. fossilis.*

It is summarized that the magnitude of $H_2O_2$ and GSH was higher in the gills than in the ARO, whereas the level of PC, TBARS, and AA, as the activity of SOD, CAT, and GPx was reduced under WDIHS. The activity of GR and respiratory complex enzymes was comparable between gills and ARO. The activity of SOD, CAT, and GPx, level of AA was higher in ARO than in gills, whereas the activity of GR and the level of the GSH was reduced under WDIHS. The activities of SOD and GPx in gills and the activity of CAT, GPx, and GR were reduced in ARO under WDIHS. The activity of SOD in ARO and CAT in gills was elevated under WDIHS. The level of AA in ARO was augmented under WDIHS. The GSH level in the gills and ARO and AA levels in the gills negatively correlated with WDIHS. The activity of the complex II enzyme had a positive correlation, whereas the rest of the mitochondrial respiratory enzymes had a negative correlation with the WDIHS (Fig. 4). It is a well-known fact that starvation may affect the expression and activity of antioxidant enzymes (*Dar et al., 2019*), and could be a factor for potential stressors, but this factor could be nullified by comparing the data set of the experimental fish with the control fish. Because both the control and experimental fish were withdrawn from food 24 h prior to the experiment. The differential activity or level of the studied redox regulatory and electron transport enzymes, ROS, oxidative stress and small antioxidant markers indicate that the fish have no sufficient defense mechanism to protect its respiring tissues *i.e.,* ARO and gills under aerial exposure induced limited $O_2$ supply condition.

## CONCLUSION

The effects of water deprivation (0, 3, 6, 12, and 18 h) induced hypoxia on the generation of ROS ($H_2O_2$), OS, activities of redox regulatory enzymes, levels of non-enzyme antioxidants, and activity of respiratory complex enzymes in respiratory organs such as ARO and gills of the catfish *Heteropneustes fossilis* was investigated for the first time. Discriminant function analyses indicate a clear contribution of the variables to influence the physiology of the fish under WDIHS. Results indicate that the ARO of the fish clearly experiences more OS than gills as this organ is employed by the fish actively under WDIHS to respire $O_2$ from the air. WDIHS induces OS in both ARO and gills, even though ARO and gills adapt

distinct OS physiological responses, especially distinct responses of specific redox regulatory molecules. The diminished functionality of ETC enzymes, specifically complexes I and III, in both organs is a plausible factor contributing to the generation of ROS. Consequently, this phenomenon may induce metabolic depression in fish subjected to WDIHS. Further research is required prior to formulating definitive conclusions regarding metabolic depression in fish that have been exposed to WDIHS.

### Funding
The work was supported by the funding to Biswaranjan Paital from the Science and Engineering Research Board, Department of Science and Technology, Govt. of India New Delhi, India (No. ECR/2016/001984) and the Department of Science and Technology, Government of Odisha (Grant letter number 1188/ST, Bhubaneswar, dated 01.03.17, ST-(Bio)-02/2017). The funders had no role in study design, data collection and analysis, decision to publish, or preparation of the manuscript.

### Grant Disclosures
The following grant information was disclosed by the authors:
The funding to Biswaranjan Paital from the Science and Engineering Research Board, Department of Science and Technology, Govt. of India New Delhi, India: ECR/2016/001984. Department of Science and Technology, Government of Odisha: Grant letter number 1188/ST, Bhubaneswar, dated 01.03.17, ST-(Bio)-02/2017.

### Competing Interests
The authors declare there are no competing interests.

### Author Contributions
- Samar Gourav Pati performed the experiments, analyzed the data, prepared figures and/or tables, and approved the final draft.
- Falguni Panda performed the experiments, authored or reviewed drafts of the article, and approved the final draft.
- Abhipsa Bal performed the experiments, authored or reviewed drafts of the article, and approved the final draft.
- Biswaranjan Paital conceived and designed the experiments, performed the experiments, analyzed the data, prepared figures and/or tables, authored or reviewed drafts of the article, and approved the final draft.
- Dipak Kumar Sahoo analyzed the data, prepared figures and/or tables, and approved the final draft.

### Animal Ethics
The following information was supplied relating to ethical approvals (i.e., approving body and any reference numbers):
The rules and regulation of the Institutional Animal Ethical committee of Odisha University of Agriculture and Technology was followed to handle the fish.

## Field Study Permissions

The following information was supplied relating to field study approvals (i.e., approving body and any reference numbers):

Not required as the animal model used is an edible fish.

## Data Availability

The raw data are available in the Supplemental File.

## Supplemental Information

Supplemental information for this article can be found online at http://dx.doi.org/10.7717/peerj.16793#supplemental-information.

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
