# Peer review of "Water deprivation-induced hypoxia and oxidative stress physiology responses in respiratory organs of the Indian stinging fish in near coastal zones"

_PeerJ, doi:10.7717/peerj.16793_

## Round 0.1 · original submission · Major Revisions

Please consider the reviewer's comments and submit an amended version with a detailed rebuttal letter.

Reviewer 1 ·

Basic reporting

1) The information provided in the introduction is sufficient to set the context for their research. However, the manuscript needs to undergo thorough revision for grammatical and syntax errors to make it easy to read and understand. For example:

a. There are lack of cohesion between some sentences. For example: "Water deprivation-induced hypoxia stress (AEHS) is studied in many fishes because it restricts water and O2 supply to their respiratory organs. And the Indian stinging fish Heteropneustes fossilis is proposed to be a future model fish in near coastal zones due to its salinity resistance capacity.". Those two sentences are not clearly connected, I suggest to use transitional expressions between sentences to show how ideas are related.

b. There are grammatical errors that need to be revised: “it” needs to be changed by “its” in line 30. If ARO is being used as a singular noun in line 58, use “experiences” instead of “experience”, etc.

c. Look for and correct any typos or spelling mistakes. For instance, “Protein” instead “protein” in line 39; line 304: "enzymes enzyme".

d. Some sentences can be rewritten to make them clearer. For example: “The role of the small redox regulatory molecules and activity of ETC although was compromised but was not sufficient to ameliorate the AEHS induced stress.” could be rewritten as: "The role of the small redox regulatory molecules and activity of ETC, although compromised, was not sufficient to ameliorate the AEHS induced stress". A general suggestion is to use comma to set off parenthetical elements.

e. Write abbreviations in their entirety before use them for the first time. For example, ARO was used for the first time in line 31, but its meaning is defined in line 35, AA was used for the first time in line 47, but it is not indicated before line 47.

f. Use more concise language by avoiding wordy sentences. For instance, “in order to” can be reduced to "to". "In general" and "in particular" appear multiple times along the manuscript and they are not always necessary to make the sentences understandable.

g. Avoid being vague and be as specific as possible. For example: in lines 66 and 74, "inhabitants" of where?; in line 94, "sufficient" for what?

h. There are long paragraphs. For instance, one of them goes from line 362-398. Long paragraphs and sentences could be cognitively difficult to digest.

2. The hypothesis needs to be rewritten to make it clearer. As a suggestion, the authors can use active voice to make it easy to follow. Moreover, it appears in the middle of the introduction; to my opinion, it should be in the last paragraph of the introduction.

3. To my opinion, the information in the lines 137-143 should precede the sentence that states the aim of the study.

4. In general, references are sufficient. However, some of them are missing, such as "Torrens 2003". On the other hand, more citation is needed in the paragraphs 362-371 and 407-419.

5. Raw data is shared, and the article has the standard structure, including figures and tables.

6. The submission is similar to another one published by the authors in 2022 (https://doi.org/10.1016/j.cbpc.2022.109300). However, in this paper they work with respiratory organs instead of liver. Moreover, the introduction, results and discussion were aborded in a different way. Therefore, this article represents an appropriate "unit of publication".

Experimental design

1. This manuscript has original primary research that fits the aims and scope of PeerJ.
2. The research question is implicit in the manuscript; however, it was not explicitly defined.
3. The research was conducted in conformity with the prevailing ethical standards in the field.
4. Methods:
a) It is not clear if the authors used biological replicates. Please, provide more information about this.
b) It is not necessary to explain that the corresponding and first authors were fully aware during the entire experimental protocol.
c) How did authors check that they isolated intact mitochondria? Did they check oxygen consumption? Since, this is important for the mitochondrial complex enzyme assay, please, provide more details (data) about this.
d) The succinate dehydrogenase enzyme participates in the Krebs cycle (KC) and in the electron transport chain (ETC). How do the authors know that the increased activity of the complex 2 comes from activity in the ETC and not from activity in the KC?
e) Even when they appear in the methodology, it is necessary to mention the units of measurement in tables 1-4.

Validity of the findings

1. There are many significant results in this paper, so benefit to literature is clear. However, there are comments that need to be answered to make sure about the validity of the findings. Mainly, the intact isolation of mitochondria and the increased activity of the succinate dehydrogenase enzyme.

2. In figure 1 is not possible to validate some of the results mentioned in the figure caption:
a) Histograms.
b) % values between histograms.
c) Error bars of standard deviation.
d) Significance (p < 0.05 level).

3. The data on which the conclusions are based are available within the article and its supplementary material.

Reviewer 2 ·

Basic reporting

Dear Editor:

The manuscript needs extensive revision for language and grammar. In its actual form, it is time-consuming. As you may notice, substantial time was spent evaluating and suggesting improvements in the introduction and materials and methods sections. I could not understand what the authors were trying to communicate through several parts of the manuscript. In my opinion, reviewers must assess research validity, significance, and originality before publication in a scientific journal. However, the many punctuation, sentence structure, grammar, and syntax errors interfere with understanding. I suggest editing help from someone with "full professional proficiency in English.

I will be thrilled to review this submission once the authors guarantee that the manuscript has been written in correct English.

Experimental design

No

Validity of the findings

No

Additional comments

No

Annotated reviews are not available for download in order to protect the identity of reviewers who chose to remain anonymous.

---

## Round 0.2 · Major Revisions

Please consider the reviewers' detailed feedback and send a revised version along with a rebuttal letter.

**Language Note:** The review process has identified that the English language must be improved. PeerJ can provide language editing services - please contact us at copyediting@peerj.com for pricing (be sure to provide your manuscript number and title). Alternatively, you should make your own arrangements to improve the language quality and provide details in your response letter. – PeerJ Staff

Reviewer 1 ·

Basic reporting

no comment

Experimental design

no comment

Validity of the findings

no comment

Additional comments

The manuscript has been substantially improved taking into account the advice given in the last version of the manuscript. Minor details to be reviewed:

Line 36-37. “...on H2O2 (hydrogen peroxide) as ROS…” instead of “on (hydrogen peroxide) H2O2 as ROS…”
Line 37. “ROS” - write the full name in the first instance and follow it immediately by the abbreviation enclosed in parentheses.
Line 43. “ROS” should be written out in full on line 37.
Lines 45 and 139. Is “II-III” correct? Or did you mean “III”?
Line 49. “Reduced glutathione” was already written out in full on line 46.
Line 50. “Ascorbic acid” was already written out in full on line 46.
Line 105. “ETC” should be written out in full on line 96.
Line 207. “SDH” should be written out in full.
Line 254. “-0.988” - it appears as a positive value (0.988) in figure 1.
Line 314. “GSH-level” instead of “SH-level”
Line 426. “mRNA” instead of “m-RNA”
Lines 428-429. “P. annectensin” instead of “Protopterus annectensin” (the full name was written out in full on line 427; so, in subsequent references you can use the first letter of the genus followed by a period and the specific epithet).
Line 441. “C. carpio” instead of “Cyprinus carpio”
Line 465. “P. annectens” instead of “Protopterus annectens”
Line 486. “3 h” instead of “3h”
Line 509. "Clarias batrachus” instead of “C. batrachus”
Figures and tables. In some of the titles of the figures and tables the species name appears as H. fossilis and in others as Heteropneustes fossilis, I suggest homogenizing the spelling.

Reviewer 2 ·

Basic reporting

.

Experimental design

.

Validity of the findings

.

Additional comments

PeerJ
Manuscript ID: 85321

Title: Water deprivation-induced hypoxia and oxidative stress physiology responses in respiratory organs of the Indian stinging fish in near coastal zones.

Authors: Samar Gourav Pati, Falguni Panda, Abhipsa Bal, Biswaranjan Paital, Dipak Kumar Sahoo.

In the current manuscript, the authors studied the response of gills and the accessory respiratory organ (ARO) of the Indian stinging fish (Heteropneustes fossilis) against the stress induced by water-deprivation hypoxia. Interestingly, according to the authors, ARO, an organ that overcomes the respiratory function of gills while this species is exposed to oxygen-deficient water or during aerial respiration, is insufficient since it lacks an effective system against the oxidative stress induced under water deprivation-induced hypoxia.
Below, I provide some remarks that authors should address and may help to improve the manuscript.

MAJOR REVISIONS
1. It is hard to believe that the manuscript was edited by a colleague, as stated by the authors, since it is plagued of grammatical errors. Authors should understand that papers can lose their impact by poor spelling and grammar. Furthermore, a poorly written paper may lead readers (including the reviewers) to put the paper aside as it might result in boring. I highlighted several grammatical and spelling errors in the PDF document that should be fixed if authors want to resubmit their manuscript.
2. On page 8 (lines 66-67), the authors mentioned that “…Depletion or fluctuation of oxygen (O2) concentration in aquatic environments is prevalent.” Are there any references supporting this? I am not sure that this could be generalized. If there is a specific O2 concentration threshold in aquatic environments, please mention it.
3. On page 9 (lines 100-101), the authors mentioned that “…To compensate for cellular energy homeostasis under a stressed state, mitochondria complex enzymes get up-regulated to transport electrons faster.” This statement may lead readers to confusion. These enzymes are not up-regulated to increase the speed of the transport of electrons (enzymes have a defined VELOCITY, which depends on the amount of substrate) but to increase the amount of transferred electrons. So, please modify this sentence.
4. In several parts of the manuscript, the authors use the term “ARO organ.” ARO is the abbreviation for accessory respiratory organ. So, please avoid using the word “organ” as it is redundant.
5. According to the authors, “… Food was withdrawn 24 h before the set-up of the experiment (page 12, lines 188-189).” So, the organisms were also fasted. I will return to this later.
6. On page 13 (line 200), the authors mentioned that “… Gills and ARO were… blotted.” What do the authors mean by blotted?
7. On page 14 (lines 229-230), the authors mentioned that “… After sonication, the mitochondrial (n = 5 in duplicates) fraction was concentrated to ~1 mg of protein in 100 µL volume.” Could authors describe how these fractions were concentrated?
8. Now, fishes were fasted. However, this issue is only considered when the authors provide an explanation for the results regarding the concentration of ascorbic acid (AA) in gills and ARO. However, even though it is well known that starvation may affect the expression and activity of antioxidant enzymes (read and discuss Dar et al., 2019), this fact is not considered a potential stressor factor.
9. On lines 387-388, the authors state that “… ARO is insufficient to compensate limited or no O2 supply”, but then they express that “… Mostly WDS-induced hypoxia regulates oxidative stress physiology more vividly in ARO than in gills.” So, are the authors suggesting that ARO is not good enough as a respiratory organ, but its performance to avoid oxidative stress is better than that of gills? If so, please elaborate in more detail.
10. The discussion section is too lengthy, meandering, and verbose. Please rewrite.
11. The resolution of the figures 1, 2, and 4 could be improved. Please modify.

Given these comments and suggestions, in my opinion, this manuscript should be accepted after major revisions and re-evaluation.

Useful references
Dar SA, Srivastava PP, Varghese T, Nazir MI, Gupta S, Krishna G. 2019. Temporal changes in superoxide dismutase, catalase, and heat shock protein 70 gene expression, cortisol and antioxidant enzymes activity of Labeo rohita fingerlings subjected to starvation and refeeding. Gene. 692:94-101.

Annotated reviews are not available for download in order to protect the identity of reviewers who chose to remain anonymous.

---

## Round 0.3 · Minor Revisions

Please take into consideration the feedback provided by the reviewer. It is a valuable resource that can be used to refine and improve the quality of your manuscript.

**Language Note:** The review process has identified that the English language must be improved. PeerJ can provide language editing services - please contact us at copyediting@peerj.com for pricing (be sure to provide your manuscript number and title). Alternatively, you should make your own arrangements to improve the language quality and provide details in your response letter. – PeerJ Staff

Reviewer 1 ·

Basic reporting

no comment

Experimental design

no comment

Validity of the findings

no comment

Additional comments

Here, I present remarks that authors ought to consider, which could potentially enhance the quality of the manuscript.

1. With respect to my remarks in the last review round, one of them remains to be addressed:

Line 267: the value "-0.988" still appears as a positive value (0.988) in Figure 1, which does not make sense because in Figure 1 it is possible to observe a negative slope.

2. It is possible to understand the message of the manuscript; however, there is still grammatical and spelling errors that should be fixed. For example:

Lines 37, 320, 329: “up to” instead of “upto” (the abbreviated form "upto" is commonly used in informal contexts)

Line 89: “possibility” instead of “possibilty”

Line 113: “an important” instead of “important”

Line 278: instead of “TBARS level were” use “TBARS level was” or “TBARS levels were”

Line 287: “compared to the control” instead of “compared to control”

Line 291: “increase” instead of “increased”

Line 292: “period” instead of “periods”. “H2O2 concentrations were” or “H2O2 concentration was” instead of “H2O2 concentrations was”

Line 312: “lower (p < 0.01)” instead of “lower(p < 0.01)”

Line 328: “exposure to WDIHS” instead of “exposure of WDIHS”

Line 385: “a mystery” instead of “mystery”

Line 394: “in extracting” instead of “to extract”

Line 424: “Under WDIHS, although ARO acts to compensate for the limited or no O2 supply to the fish body through the gills” instead of “Under WDIHS, although ARO is insufficient to compensate for limited or no O2 supply to the body of fish via gills,”

Line 427: “0 h” instead of “0h”

Line 472: “exposure rather than gills”? instead of “exposure than gills”

Line 501: “titer”? instead of “titre”

Line 528: “are supported by the findings” instead of “are supported with the findings”

Line 571: “gills” instead of “gils”

Line 594: “fulfills” instead of “fulfils”

Line 601: “a distinct overlapping among 6, 12 and 18 h groups indicates” instead of “a distinct overlapping among 6, 12 and 18 h groups indicate”

Line 617: “by comparing” instead of “with comparing”

---

## Round 0.4 · accepted · Accept

Thanks for addressing the revisions requested. Now, your manuscript is accepted in PeerJ.